# p53’s Extended Reach: The Mutant p53 Secretome

**DOI:** 10.3390/biom10020307

**Published:** 2020-02-15

**Authors:** Evangelos Pavlakis, Thorsten Stiewe

**Affiliations:** Institute of Molecular Oncology, Member of the German Center for Lung Research (DZL), Philipps-University, Hans-Meerwein-Straße 3, 35043 Marburg, Germany; evangelos.pavlakis@staff.uni-marburg.de

**Keywords:** p53, tumor suppressor, secretome, extracellular vesicles, exosomes, tumor microenvironment, pre-metastatic niches, metastasis

## Abstract

p53 suppresses tumorigenesis by activating a plethora of effector pathways. While most of these operate primarily inside of cells to limit proliferation and survival of incipient cancer cells, many extend to the extracellular space. In particular, p53 controls expression and secretion of numerous extracellular factors that are either soluble or contained within extracellular vesicles such as exosomes. As part of the cellular secretome, they execute key roles in cell-cell communication and extracellular matrix remodeling. Mutations in the p53-encoding *TP53* gene are the most frequent genetic alterations in cancer cells, and therefore, have profound impact on the composition of the tumor cell secretome. In this review, we discuss how the loss or dominant-negative inhibition of wild-type p53 in concert with a gain of neomorphic properties observed for many mutant p53 proteins, shapes a tumor cell secretome that creates a supportive microenvironment at the primary tumor site and primes niches in distant organs for future metastatic colonization.

## 1. Introduction

Originally discovered at the highpoint of tumor virus research and initially classified as an oncogene, the tumor suppressor p53 is now considered one of the most critical protectors of the human genome and a central component of a multiplex molecular network of signaling cascades [1,2,3,4]. Subsequent to various cellular insults that include DNA damage, oxidative stress, and oncogenic signaling, p53 is activated and functions as a sequence-specific transcription factor, setting in motion pathways for DNA repair, cell cycle arrest, apoptosis, and senescence [4,5,6,7]. Interestingly, p53-like genes resembling mammalian p53 in sequence and function were first identified in the evolution of modern-day simple invertebrate organisms that are not inclined to tumor development [4,8,9]. Similar to mammalian p53, these genes trigger cell death upon stress stimuli; however, they are only expressed in the germline, suggesting that p53 did not evolve to prevent cancer but rather to protect cells from genomic instability and replicative stress [4,10,11]. p53’s ability to defeat somatic tumors at advanced age evolved later synchronously with the evolution of multicellular life [10,11,12].

p53’s potency in suppressing inappropriate clonal outgrowth is unparalleled and clearly evident from the severe cancer susceptibility of mice and men with engineered or inherited mutations in p53, respectively [13,14,15]. To overcome p53-mediated tumor suppression, cancer cells have developed multiple tactics to disarm p53 and, thereby, promote their own survival and expansion. The most direct way is certainly through *TP53* gene mutations, and cancer genome sequencing projects have provided undeniable evidence showing that *TP53* alterations are the most frequent events in human cancers [16,17,18]. *TP53* is now known to be hit largely by missense mutations, although deletions, truncations, and frameshift mutations have also been reported [16,18]. Among the missense mutations, roughly 80% affect residues within the p53 DNA-binding core domain, where several mutational hotspots have been acknowledged [16,18]. These missense mutants have lost their ability to bind to the established p53-responsive DNA elements and launch the respective tumor suppressive programs (loss of function, LOF). In addition, missense mutants bind and inactivate wild-type proteins expressed from a nonmutated allele (dominant-negative effect, DN), and many acquire new neomorphic activities (gain of function, GOF) that boost cancer cell growth, survival, expansion, and spread in many different ways [19,20,21,22,23]. For instance, mutant p53 has been shown to control several tumor cell-autonomous processes beneficial for tumor cell survival under adverse conditions, including regulation of energy metabolism, response to proapoptotic signals, and adaptation to oxidative stress [21,24].

Apart from these well-known functions within tumor cells, *TP53* mutations also affect how tumor cells interact with their surroundings, i.e., the various types of stroma cells in the microenvironment and the extracellular matrix in which tumor and stroma cells are embedded. The communication with the components of the tumor stroma is bi-directional and largely mediated by factors secreted by tumor cells into the extracellular space. All the secreted factors together are referred to as the tumor secretome, comprised of protein and other nonprotein molecules, including lipids or metabolites. Collectively, the tumor secretome acts to blunt tumor-suppressive activities present in the stroma and to reprogram the microenvironment into a tumor-supportive neighborhood. For the purpose of this review, we will focus on secreted proteins and discuss how *TP53* mutations affect the protein secretome of tumor cells and thereby shape the local and distant microenvironment to foster invasion, metastasis, and drive tumor progression to a more aggressive and therapy-refractory state.

## 2. *TP53* Mutations

The progress with massively parallel sequencing of tumor genomes in the past decade has provided an unprecedented insight into the numerous ways in which the *TP53* locus is altered in tumors and how this unique *TP53* “mutome” translates into functional consequences, leading ultimately to more aggressive tumorigenesis and a poor patient outcome [18,25].

### 2.1. Classes of TP53 Mutations

*TP53* mutations are dispersed throughout all exons with a striking preference for the central region encoding the DNA-binding core domain. The most common (72.7%) and well-characterized *TP53* mutations among the 80,400 cancer cases reported in the Universal Mutation Database (UMD) are missense mutations in the DNA-binding domain (DBD), signifying that DNA binding is crucial for the tumor suppressive function [16,26]. Six hotspot residues within the DBD (R175, G245, R248, R249, R273, and R282) are hit most frequently. Depending on whether the corresponding residues are involved in DNA contact or structure maintenance, mutant proteins are categorized as contact (R273H, R248Q, and R248W) or conformational (R175H, G245S, R249S, and R282H) [27,28]. Contact mutants derive from missense mutations in residues responsible for direct contact with the DNA sequences forming p53 response elements in target gene promoters and have an intact native fold [29,30,31]. Conformational mutations result in the disruption of the p53 protein structure by decreasing the already low folding stability of the DBD, leading to its denaturation and often aggregation at body temperature [27]. Nevertheless, the distinction between these two mutation categories is somewhat arbitrary, as there are p53 mutants that, in principle, fit in both (e.g., R248Q) [27,32]. In addition, there are DBD mutations that do not fit into this bipartite classification, such as cooperativity mutations which influence the formation of the DNA-bound tetramer without any alterations at the DNA-binding surface or the overall folding and stability of the DBD [33,34,35,36,37,38,39,40,41,42,43].

### 2.2. Tumor Promotion by TP53 Mutations

The notable preference for p53 missense mutations in human cancers led to the appreciation that mutant p53 proteins convey a selective advantage during tumorigenesis. On the one hand, mutant p53 can disable wild-type p53 expressed from the second allele through oligomerization (dominant-negative effect, DN) [44,45]. On the other, p53 mutants can acquire novel oncogenic functions [46]. Supporting the idea of a pro-tumorigenic gain of oncogenic activity, overexpression of mutant p53 in murine or human cells with no endogenous p53 was reported to promote malignant transformation [47,48]. Remarkably, mice carrying the R172H allele alone or in combination with wild-type p53 exhibited similar survival compared to mice homo- or heterozygous for a p53-null allele [49,50]. They did, however, present a more diverse spectrum of more aggressive and metastatic tumors, including a higher incidence of carcinomas [49,50]. In parallel, humanized p53 knock-in mice (HUPKI) where the DNA-binding domain of the p53 locus was partly substituted by the corresponding human sequence with or without hotspot mutations resulted in similar conclusions [51,52]. Notably, a recent study also reported that mouse mutant equivalents of R175H and R248W lead to rapid breast cancer development driving a parallel evolutionary pattern of metastases [53]. Consistent with these animal studies, Li-Fraumeni patients carrying *TP53* missense mutations in the germline develop cancer significantly earlier than patients with nonsense or frame-shift mutations that result in loss of p53 protein expression by nonsense-mediated mRNA decay [54,55]. Overall, these in vivo genetic studies effectively portrayed the crucial role of mutant p53 proteins in the development and metastatic progression of tumors and laid the foundations for the mutant p53 GOF concept. Of note, the GOF concept is not undisputed, and two recent large-scale functional studies of the p53 mutome have revealed the dominant-negative effect as the primary unit of selection for *TP53* missense mutations [56,57].

Even though we often refer to mutated p53 proteins as “mutant p53” (or short, “mutp53”), we are becoming increasingly aware of the incredible functional diversity within the p53 mutome [21,25]. p53 cancer mutants differ dramatically regarding the loss of wild-type activity, the degree of dominant-negative activity, and the quantity and quality of neomorphic (GOF) activities [15,22,38,58]. With respect to the numerous ways that p53 mutants operate to alter the tumor cell secretome, one may therefore differentiate between mechanisms that are caused by the abrogation or inhibition of wild-type p53 functions (loss-of-function and dominant-negative activities) or true neomorphic mutant p53 activities (Figure 1). 

## 3. Abrogation of the Wild-Type p53 Secretome (LOF, DN)

The tumor suppressive function of wild-type p53 is most commonly considered a cell-autonomous effect brought about by transactivation of antiproliferative or cytotoxic target genes. However, the p53 target gene spectrum also comprises numerous genes encoding for secreted proteins that trigger changes outside of the tumor cell [59,60]. In addition, many of the direct immediate targets of p53 induce activation of secondary programs that are not confined to the cell interior and extend to the extracellular space [61,62].

### 3.1. Senescence-Associated Secretory Phenotype

Antiproliferative target genes of p53 orchestrate the senescence program, which not only permanently arrests cell proliferation but also triggers secretion of proteins that transform the microenvironment in a non-cell-autonomous manner (Figure 2) [63,64]. The multifaceted senescence process exerts both anti- and pro-tumorigenic effects in a strongly context-dependent manner and downregulates the production of extracellular matrix (ECM) factors, upregulates enzymes that degrade the ECM, and promotes the secretion of a multitude of inflammatory cytokines and immunomodulators, collectively defined as the senescence-associated secretory phenotype (SASP) [62]. Several SASP components such as IGFBP-7, PAI-1, IL-6, IL-8, and CXCL-1/GRO1 are critical reinforcers of senescence [65,66,67,68,69]. p53 plays a crucial part in this process, as many of the senescence-related secreted factors are directly stimulated by p53, frequently co-regulated together with NF-kB [65,70]. Just as the cell-autonomous effects of senescence pose a barrier to malignant transformation, the non-cell-autonomous functions associated with SASP also limit tumorigenesis. SASP potentiates the tumor suppressive power of the senescence program by transmitting the senescent phenotype in a paracrine fashion to adjacent cells [71]. In addition, p53-induced senescent cells may restrict angiogenesis via upregulation of the anti-angiogenic protein Thrombospondin-1 (TSP-1) [72]. In the liver, p53-induced senescence is part of a homeostatic program which, under chronic damage, restricts fibrosis and enables healing, whilst modulating the secretion of cytokines produced by hepatic stellate cells (HSCs) that are in turn selectively targeted and eliminated by NK cells [73]. p53 activity in senescent HSCs also stimulates the secretion of factors that skew macrophage polarization to the tumor-fighting M1-like state capable of attacking and eliminating senescent cells [61]. Similarly, pre-malignant senescent hepatocytes secrete chemo- and cytokines that trigger immune-clearance dependent on CD4+ T cells and monocytes/macrophages [74].

These non-cell-autonomous tumor-suppressive functions of wild-type p53 are commonly lost by p53 mutants or blocked by them via the dominant-negative effect, resulting in a more tumor-supportive and less hostile microenvironment [64]. p53-deficient HSCs, for instance, fail to enter senescence, resulting in the polarization of macrophages to a tumor-promoting M2-state characterized by the secretion of nourishing factors that enhance the proliferation of premalignant cells [61]. 

Of note, although senescence is generally considered as tumor suppressive, it has also been shown to exert context-dependent pro-tumorigenic effects, e.g., via attraction of tumor-supporting inflammatory cells, such as macrophages and fibroblasts, and angiogenesis [62,75,76,77]. In a mouse model of colorectal cancer, genetic loss of CKIα (*Csnk1a1*) in intestinal epithelial cells induces a senescence-associated inflammatory response that efficiently antagonizes the proliferative activity of Wnt signaling and counteracts tumorigenesis. In the absence of p53, this senescence-associated inflammatory response is maintained but loses its growth control capacity and instead actively accelerates tumor cell growth and invasiveness, which highlights p53 status as a critical determinant of the pro- versus anti-tumorigenic activity of the senescence-associated secretome [78]. Furthermore, cells escaping from senescence were shown to display increased stemness. As the senescent phenotype is transferred from critically stressed cells via SASP to adjacent tumor cells [71], the resulting increase in their stemness may drive tumor progression to a more aggressive and drug-resistant state [79]. Importantly, inactivation of the senescence program through loss of p53 is crucial also for tumor maintenance, as reinstating wild-type p53 in p53-null hepatocellular carcinomas in vivo triggers senescence associated with upregulation of inflammatory chemokines and cytokines that drive an innate immune response leading to tumor clearance involving enhanced NKG2D-dependent tumor elimination by natural killer cells [80,81].

### 3.2. Crosstalk with NF-κB

NF-κB plays key roles in immunity and inflammation, is activated by stimuli such as pathogen-associated molecular patterns (PAMPs) and TNFα, and drives the expression of various pro-inflammatory genes to stimulate secretion of cytokines [82,83,84]. Consequently, deregulated NF-κB activation contributes to the pathogenesis of several diseases that contain an inflammatory component, including cancer. Similar to p53, the NF-κB pathway is also activated by various stress stimuli, including DNA damage, and plays central roles in the control of proliferation and apoptosis [82]. Given these similarities, it is not surprising that an integrative crosstalk between the two pathways is crucial for cell-fate decisions (Figure 3).

While the p53 response is associated with inhibition of cell cycle proliferation or induction of cell death, the NF-κB response is more variable—in most cases, antagonistic to p53 but, in some contexts, also cooperating with p53 [85]. NF-κB negatively controls p53 levels by direct transcriptional upregulation of MDM2 [86,87] and inhibits the pro-apoptotic functions of p53 via upregulation of anti-apoptotic genes [85]. Vice versa, p53 directly inhibits the ability of NF κB to transactivate target genes by interacting with p65/RelA [88,89]. Furthermore, NF-κB and p53 inhibit each other in a reciprocal manner by competing for binding to the p300 and CBP coactivators, which are required by both for efficient transactivation of target genes [90,91,92]. This competition is regulated by IKKα-mediated CBP phosphorylation, which increases histone acetyltransferase activity and shifts CBP binding from p53 to NF-κB, resulting in increased cell proliferation [93]. The in vivo role of the antagonistic p53/NF-κB crosstalk for cancer development was shown in a mouse model of Kras^G12D^-driven lung adenocarcinoma, where loss of p53 triggers NF-κB activation essential for tumor development, whereas restoration of p53 in established p53-null lung tumors leads to NF-κB inhibition and tumor suppression [94]. Similarly, in a mouse model of inflammation-induced colorectal cancer, loss of p53 during tumor progression is associated with increased intestinal permeability, causing formation of an NF-κB-dependent inflammatory microenvironment and the induction of epithelial-mesenchymal transition [95]. Furthermore, consistent with suppression of the transactivation ability of NF-κB by p53, p53-null mice show increased NF-κB activity, are hypersensitive to LPS-induced septic shock, and display increased macrophage recruitment and delayed neutrophil clearance in response to inflammation-inducing agents [96].

However, in other settings, p53 cooperates with NF-κB. For example, induction of apoptosis by p53 was shown to require activation of NF-κB, possibly because p53-mediated transactivation of the pro-apoptotic target genes *Noxa* and *p53AIP1* is dependent on NF-κB [97,98]. In normal macrophages, p53 and NF-kB co-activation results in induction of pro-inflammatory cytokines and chemokines like CXCL-1, IL-6, and IL-8, resulting in neutrophil recruitment and inflammation [99]. In tumor-conditioned macrophages as a model for tumor-associated macrophages (TAMs), p53 is constitutively activated in response to paracrine signals from tumor cells and facilitates secretion of IL-6, which has well-documented pro-tumor effects by promoting tumor cell survival, angiogenesis, and recruitment of tumor-associated neutrophils (TAN) [99]. Interestingly, the cooperative transactivation of inflammatory target genes by p53 and NF-κB is modified by a common *TP53* polymorphism at codon 72, as the P72 variant interacts better with the p65 RelA subunit of NF-κB than the more prevalent R72 variant [100]. P72 knock-in mice therefore show an enhanced response to inflammatory challenge [100].

In light of these observations, it is not surprising that the loss of p53 wild-type activity caused by most *TP53* mutations results in a deregulated tumor cell secretome impacting on the reciprocal crosstalk of tumor cells with the plethora of cell types in their microenvironment. In a Kras^G12D^-driven mouse model of pancreatic cancer, loss of p53 promotes recruitment and instruction of suppressive myeloid CD11b^+^ cells, in part through increased expression of CXCR3/CCR2-associated chemokines and macrophage colony-stimulating factor (M-CSF) [101]. In addition, the p53-null pancreatic tumors show an accumulation of suppressive regulatory T (Treg) cells that cooperate with suppressive myeloid cells to blunt anti-tumor CD4^+^ T helper 1 (Th1) and CD8^+^ T cell responses [101]. Likewise, in a mouse study of PTEN-driven prostate cancer loss of p53 enhances tumor infiltration of CD11b^+^Gr1^+^ polymorphonuclear cells through increased CXCL17 secretion and expansion of immunosuppressive Treg cells [102]. In breast cancer cells, loss of p53 results in increased WNT secretion that stimulates tumor-associated macrophages to produce IL-1β, thus driving systemic neutrophil inflammation and metastasis [103]. The extent to which loss of wild-type p53 alters the secretome is also underlined by the profound secretory changes observed upon reconstitution of wild-type p53 in p53-null tumor cells. Comprehensive proteomic analyses revealed at least 50 secreted proteins to be differentially controlled by p53 [104]. Of these proteins, several have well-defined tumor-related functions in ECM modulation (MMP1, MMP13, TIMP-3, and MMP2); tumor cell survival (BDNF, FGF-4, and IGFBPs); regulation of immune responses (B-2M, IL-8, and attractin); cell-cell communication (α-catenin and β-5 tubulin); angiogenesis (VEGF, PEDF, and CYR61); and the cell migration-invasion-metastasis cascade (ADAM-10, TGF-β, Tau, and MMPs) [104]. Along the same line, treatment of hepatocellular carcinoma cells with Mdm2 inhibitors activated p53-mediated gene expression and protein secretion of steroid hormone binding factors, such as SHBG (sex hormone-binding globulin) and CBG (corticosteroid-binding globulin), capable of promoting apoptosis in breast cancer cells [105]. 

Together these examples illustrate the tremendous tumor suppressive power of the wild-type p53 secretome and the consequences resulting from its loss due to p53 mutations.

## 4. Mutant p53 GOF Mechanisms Altering the Tumor Secretome

Although most p53 mutations largely abrogate the DNA binding activity of p53, directly or in a dominant-negative fashion, mutant p53 proteins are far from transcriptionally inert and can modulate gene expression indirectly by binding and regulating numerous transcription factors and chromatin remodelers (Figure 4) [19,106]. Furthermore, p53 mutants can interact with cellular proteins involved in signal transduction and secretory pathways resulting in altered protein secretion.

### 4.1. Mutant p53 Modifies the Activity of Other TFs

Although there is some evidence that mutant p53—similar to wild-type p53—binds DNA in a sequence-unspecific but structure-specific manner [107,108,109,110], mutant p53 seems to interact with DNA primarily indirectly through physical association with other transcription factors that recruit mutant p53 to their respective target genes [19,22,24,111]. Although many of the TFs bound by mutant p53 also interact with wild-type p53, the functional outcome is often opposite. As these mutant p53 activities are dependent on other transcription factors, many of which are expressed in a cell- or stimulus-specific manner, there is considerable context-specificity. In addition, the interactions between mutant p53 and other TFs are structurally quite diverse, and therefore, also influenced by the type of mutation (conformational/contact) or the modified residue.

#### 4.1.1. p53 Family Members

One of the best-studied mechanisms by which p53 mutants impact on gene expression is via co-aggregation and sequestration of the two tumor suppressive family members p63 and p73 [49,112,113,114]. While p63 and p73 have been demonstrated to form homo- and heterotetramers with each other, neither forms heterotetramers with wild-type p53 [115]. Instead, mutant p53 interacts with the family members through the DNA-binding core domain [113,116]. Specific interactions between mutant p53 and p63/p73 may result from the structural changes in the DNA-binding core domain that are characteristic for missense mutations of the conformational class, but p53 contact mutants have also been shown to interact with p63 and p73 [117,118]. Given that p63 and p73 are sequence-specific transcription factors, interaction with p53 mutants alters their activity at their target genes positively or negatively depending on promoter context [113]. Both activation and repression have been demonstrated to increase invasion and metastasis. For example, certain p53 mutants but not wild-type p53 may interact and inhibit the family member TAp63 to control pro-invasive and pro-metastatic transcriptional profiles via regulation of Sharp1, Cyclin G2, and miR-155 [113,116,119,120,121]. Mutp53-p63 interaction also drives invasion and metastasis by stimulating TGF-β signaling, recycling of EGFR and integrins via RCP, and downregulating Dicer-mediated processing of anti-metastatic miRNAs [24,116,120,122,123]. Importantly, mutant p53 can use p63 as a molecular chaperone to tether to its target gene’s promoters and alter gene expression profiles to promote oncogenesis [124]. Expression profiling of six different inducible mutant p53 cell lines revealed a core set of 59 target genes highly enriched with p63 target genes that encode secreted protein products and form a pro-invasive secretome [124].

Furthermore, quantitative proteomic characterization of the secretome from non-small cell lung cancer cells has demonstrated the power of mutant p53 to drive expression of secreted proteins that function in either autocrine or paracrine signaling to promote migration and invasion of tumor cells [125]. Among them were BIGH3, an ECM protein that modulates cell adhesion, alpha-1 antitrypsin (A1AT or SERPINA1), a secreted serine protease inhibitor, and various epithelial-mesenchymal transition markers. A1AT was upregulated in a transcriptional manner via cooperation of mutant p53 with the family member p63 and identified as a critical and indispensable target for cell migration and invasion in vitro and in vivo [125]. A1AT expression correlated with increased tumor stage and shorter survival of lung cancer patients and was predominant at the boundary between tumor and stroma, suggesting a primary function in tumor-stroma crosstalk [125]. Additionally, in several other human cancer types, A1AT expression was shown to correlate with invasion and metastasis [126,127,128], and high levels of secreted A1AT in blood and urine samples from cancer patients were associated with worse patient outcomes [126,129]. Although the pleiotropic mechanisms of A1AT in tumorigenesis are not fully understood, A1AT presumably functions at least in part via inhibiting the activity of enzymes present in the extracellular environment. For example, inhibition of proteases, like mast cell chymase and leukocyte elastase, can influence TGF-β availability in the ECM, and thereby, affect tumor cell invasion [130]. In addition, A1AT has been reported to induce the release of the angiopoietin-like protein 4 via a peroxisome proliferator-activated receptor-dependent pathway [131], indicating that protease inhibitor-independent functions likely contribute to its activity as a mediator of mutant p53’s pro-tumorigenic activity.

#### 4.1.2. NF-κB

Other examples include complexes of mutant p53 with the TFs NF-κB, NF-Y, Ets-1 and Ets-2, SP1, VDR, E2F1, ID4, and SREBP [22]. Apart from the crosstalk of NF-κB and wild-type p53 described in the previous section, p53 mutations have been shown to modulate NF-κB-driven inflammatory gene expression in multiple ways independent of their effect on wild-type p53. First, mutant p53 was shown to enhance induction of NF-κB activity by TNF-α associated with enhanced nuclear accumulation of NF-κB [132]. Similarly, certain p53 mutants augment NF-κB activity, resulting in increased CXCL5, CXCL8, and CXCL12 chemokine expression crucial for tumor cell migration [133]. Interestingly, CXCL8 itself has the capacity to trigger NF-kB activation, suggesting that mutant p53 might initiate the formation of an autocrine or paracrine feedback loop [134]. Furthermore, wide transcriptome profiling of various cancer cell lines revealed a cancer-related gene signature (CGS) encoding mainly pro-cancerous secreted molecules that is synergistically upregulated by p53 inactivation and oncogenic Hras^V12^ expression [135,136]. The identified CGS comprises many ECM-related proteins, like TFPI2, MMP3, PRSS2, C1QTNF1, and ADAMTS8, but also various CXC chemokines and cytokines [136]. Interestingly, various p53 mutations were found to promote the secretion of CGS factors via distinct mechanisms and to different extents. Loss of wild-type p53 reduces expression of the p53 target gene BTG2, which normally represses Hras by reducing its GTP-loaded state [135]. The zinc region conformational mutants R175H and R179R also augment Hras activity by inhibiting BTG2, while contact mutants R248Q and R273H induce a significantly higher CGS expression by boosting NF-κB activation [137]. Notably, the L3 loop conformation mutant G245S does not seem to affect the CGS, which might be explained by the observation that this mutation partially retains some wild-type p53 characteristics [137]. In a mouse model for colorectal cancer induced by chronic inflammation, mutant p53—but not loss of p53—triggered a pronounced NF-κB-mediated inflammatory response acting as a driver of colorectal tumorigenesis and invasion [138].

Mechanistically, early chromatin immunoprecipitation (ChIP-on-Chip) studies revealed an overrepresentation of NF-κB binding sites among mutant p53-bound sequences [139]. Later, in-depth ChIP-seq coupled with transcriptome (RNA-seq and GRO-seq) analyses demonstrated a global overlap in the binding of p53 mutants and NF-κB/p65 that drives alterations in enhancer and gene activation in response to chronic TNF-α signaling [140]. p53 mutants were shown to directly interact with NF-κB/p65 impacting on each other’s DNA binding profile [140]. Simultaneous and cooperative binding of mutant p53 and NF-κB was shown to be required for RNA polymerase II recruitment, synthesis of enhancer RNAs, and activation of various tumor-promoting secreted factors, including MMP9 and CCL2 [140]. Of note, mutant p53 was also shown to directly bind and activate the promoters of the NF-κB target genes CXCL-1 (GRO1) without changes in NF-κB signaling, presenting an alternative NF-kB-independent mechanism for the induction of cytokine secretion [141].

Apart from these nuclear functions of mutant p53, cytoplasmic activities were also shown to impinge on inflammatory NF-κB signaling. In response to TNF-α, mutant p53 sustains activation of NF-κB while dampening activation of JNK by directly binding and inactivating the cytoplasmic RasGAP DAB2IP, a tumor-suppressive protein that switches the TNF-α response from NF-κB to JNK signaling [142]. As a consequence, the mutp53-DAB2IP interaction enhances the invasive behavior of cancer cells exposed to an inflammatory microenvironment and represents an indirect means of how neomorphic properties of mutant p53 enhance pro-tumorigenic NF-κB signaling [142].

#### 4.1.3. STAT Transcription Factors

In a mouse model of chronic colitis-associated colorectal cancer, p53 mutants were found to interact with the signal-transducing STAT3 TF and enhance its transcriptional activity by displacing the STAT3-inhibitory SHP2 phosphatase [143]. This mutp53/STAT3 axis was shown to be crucial for colorectal cancer cell survival and a pro-invasive epithelial-to-mesenchymal transition (EMT) phenotype [143]. Intriguingly, mutp53-activated STAT3 signaling occurred in the absence of an NF-κB-driven secretory response observed in different colitis-associated colorectal cancer models [95,138,143]. It can be speculated that the distinct microenvironmental phenotypes reflect differences in experimental protocols (acute/transient versus chronic inflammation) and p53 mutation status (null versus mutant alleles and global versus tissue-specific mutations), further illustrating the highly context-specific effects of p53 mutations on the tumor secretome.

Recent studies have also revealed a connection of mutant p53 to STAT1 signaling in the context of matricellular protein secretion [144]. Matricellular proteins are distinct from the fibrillar ECM proteins with primary structural roles and have emerged as important biological mediators of cell function by promoting ECM remodeling and initiating downstream signaling via integrins or RTKs [145]. The matricellular protein periostin (POSTN) was shown to be important for cancer cell dissemination and metastatic colonization, and, accordingly, POSTN overexpression has been observed in numerous advanced stage cancers, including breast and NSCLC, where the p53 mutation rate is high [144]. In a study on esophageal cancer, the p53 missense mutant R175H was shown to cooperate with POSTN to enhance STAT1 signaling [144]. The STAT1-regulated gene network includes numerous secreted factors known to promote tumor progression via maintenance of a pro-inflammatory microenvironment, suggesting that mutp53-POSTN cooperation contributes to a permissive tumor microenvironment that facilitates invasion.

#### 4.1.4. SP1

Furthermore, p53 mutants interact with the SP1 transcription factor at the EGFR promoter, resulting in promoter activation by histone acetyltransferase recruitment [146]. In addition, mutp53/SP1 complexes also mediate transactivation of the ectonucleoside triphosphate diphosphohydrolase 5 (ENTPD5) gene, which encodes an endoplasmic reticulum (ER)-resident UDPase [147,148,149,150]. By cleaving UDP to UMP in the ER, ENTPD5 promotes import of UDP-glucose into the ER, which is essential for the proper folding of nascent N-glycoproteins by the calnexin/calreticulin (CANX/CALR) chaperone cycle. Since the majority of secreted and membrane-bound proteins are N-glycoproteins, and therefore, strongly depend on processing by the CANX/CALR cycle, upregulation of the N-glycoprotein-folding capacity by mutant p53 can be expected to heavily boost protein secretion [151,152]. In addition, upregulation of CANX/CALR cycle activity modifies the composition of the secretome, as various N-glycoproteins rely on chaperones to a different extent [151,152,153]. While it was shown that mutp53-induced ENTPD5 expression is crucial for cell migration, invasion, and lung colonization by p53-mutant breast cancer cells, the relevant N-glycoproteins affected by the mutp53-ENTPD5 axis remain unknown.

#### 4.1.5. E2F1

The p53 mutants R175H, R273H, and R280K were shown to direct a VEGF-independent transcriptional program of tumor neo-angiogenesis via upregulation of an inhibitor of DNA-binding 4 (ID4), involving assembly of mutp53-E2F1 complexes on regulatory regions of the ID4 promoter [154]. ID4, in turn, promotes the expression and secretion of pro-angiogenic factors, like the cytokines IL8 and GRO-α, resulting in increased endothelial cell proliferation and migration [154]. Furthermore, in complexes with ID4, mutant p53 controls VEGFA isoforms by recruiting the long noncoding RNA MALAT1—itself a critical regulator of metastasis [155,156]. Interestingly, wild-type p53 expression has the opposite effect; under hypoxic conditions, wild-type p53 associates with E2F1 to suppress VEGF expression and inhibit angiogenesis [154]. In mouse fibroblasts, mutant p53 overexpression induces activation of protein kinase C and, in turn, augments the expression of vascular endothelial growth factor (VEGF) [157]. Similarly, in a model of pre-B acute lymphoblastic leukemia, mutant p53 expression in bone marrow stromal cells leads to increased synthesis and secretion of VEGF into the tumor stroma and stimulates growth of leukemic cells via both autocrine and paracrine mechanisms [158]. VEGF expression is also related to the p53 status in human breast cancer patients where higher VEGF expression, observed when p53 is mutated, reflects a worse patient outcome [159].

#### 4.1.6. HIF1

Mutant p53 was also shown to interact with the hypoxia-inducible transcription factor (HIF1) [59]. Composition of the tumor microenvironment is strongly influenced by oxygen availability and subject to control by the hypoxia signaling network. As tumors grow, adaptation to lower oxygen levels is a vital step for progression towards more advanced stages [160,161]. Due to this hypoxic microenvironment, cancer cells greatly depend on processes stimulated by HIF1, which, under normoxic conditions, is continuously degraded but protected from the proteasome when oxygen is missing. HIF1 manages a complex transcriptional program that triggers angiogenesis, metabolic alterations, invasiveness, and metastatic dissemination, thus driving malignant progression and resulting in poor patient outcomes [162]. Under conditions of hypoxia, p53 mutants accumulating in late-stage non-small cell lung cancer (NSCLC) were found to assemble into a complex with HIF1, which binds to the SWI/SNF chromatin remodeling complex to promote the expression of a subset of hypoxia-responsive HIF1 target genes [163]. Importantly, depletion of mutant p53 impairs the HIF1-mediated up-regulation of various ECM components, including type VIIa1 collagen and laminin-γ2, and reduces the tumorigenic potential of NSCLC cells in vitro and in mouse models in vivo [163]. Mutant p53 interactions with HIF1 therefore boost hypoxia-induced stroma remodeling and thereby contribute to an invasive histological pattern which is a characteristic hallmark of advanced tumors [163].

#### 4.1.7. Other TFs

Chromatin immunoprecipitation analysis of mutant p53 by ChIP-on-Chip also revealed a significant overrepresentation of vitamin D receptor (VDR) binding sites at mutp53-bound loci [164]. By increasing nuclear VDR levels and directly interacting with the VDR transcription factor, mutant p53 modulates the vitamin D3-induced transcription of numerous vitamin D-responsive genes, including secreted factors such as IGFBP3, and converts vitamin D into an anti-apoptotic agent [164]. In addition, mutant p53 is interacting with the NF-Y transcription factor and recruiting the p300 histone acetyltransferase to cell cycle genes with NF-Y-binding CCAAT boxes [165,166]. While the mutp53/NF-Y complex has clear cell-autonomous roles in cell cycle control, non-cell-autonomous functions have not yet been described.

### 4.2. Mutant p53 Modifies the Activity of Chromatin-Regulators

Another tactic by which p53 mutations control the tumor secretome at the transcriptome level is through modulation of the chromatin landscape. For example, p53 mutant tumor cells show an elevated expression of the chromatin-regulatory COMPASS (complex proteins associated with Set1) complex subunits MLL1 (KMT2A), MML2 (KMT2D), and MOZ (KAT6A), which possess histone methyl- and acetyl-transferase activities, respectively [167]. Similarly, as in the previous section, mutant p53-mediated upregulation of these enzymatic activities was shown to involve mutp53-Ets-2 interactions for recruitment of the mutated p53 protein to the MLL1, MLL2, and MOZ gene promoters. COMPASS inhibitors selectively compromise growth of p53-mutated cells, indicating a specific requirement for mutp53-dependent chromatin regulation for the maintenance of a highly malignant phenotype. Although it is known that the COMPASS complex is critical for the expression of numerous inflammatory target genes [168], the impact of the mutp53-COMPASS axis on the tumor secretome remains to be investigated.

In addition, mutant p53 regulates expression of numerous genes in a manner dependent on the SWI/SNF nucleosome remodeling complex [163,169]. As described above, the SWI/SNF complex can be recruited to genes by a mutp53/HIF1 complex assembled in hypoxic tumor cells [163]. Vice versa, the SWI/SNF complex can recruit mutant p53 to promoters, as shown for VEGFR2 [169]. This results in the remodeling of the promoter, the maintenance of an open conformation, and the transcriptional activation of the corresponding VEGFR2 gene [169]. Genome-wide expression profiling revealed that roughly 50% of the genes regulated by mutant p53 were also regulated by SWI/SNF, indicating that mutant p53 harnesses SWI/SNF to remodel a vast variety of promoters into a transcriptionally active conformation, including important secreted factors such as IGFBP5, ceruloplasmin, and mammaglobin-A [169]

### 4.3. Other Transcriptional Mechanisms

Mutant p53 was described to enhance interleukin-1 (IL-1) signaling by repressing the promoter of the secreted IL-1 receptor antagonist sIL-1Ra [170]. Cytokines such as interleukins, interferons, or TNF-α function as molecular messengers, allow immune cells to communicate with one another, and are involved in tumor immune-surveillance in an efficiently regulated multifaceted, pleiotropic, and redundant manner [171]. IL-1, for example, is secreted by stromal cells and infiltrating leukocytes during inflammation and is intrinsically and extrinsically involved in cancer pathology [172]. Antagonists of IL-1 bind to IL-1 receptors without conveying any stimulating signals, thus blocking the pro-inflammatory signaling by IL-1 [173]. Mutant p53 physically interacts with the sIL-1Ra promoter and, together with the transcriptional co-repressor MAFF (v-MAF musculoaponeurotic fibrosarcoma oncogene family, protein F), represses gene transcription [170]. Thereby, mutant p53 contributes to a pro-inflammatory tumor microenvironment and sustains IL-1-driven tumor malignancy [170]. 

Mutant p53 also controls the activity of matrix metalloproteinases (MMPs), which play a critical role in cancer cell invasion by degrading several ECM constituents [174,175]. In tumor types, where p53 mutants accumulate to high levels, expression of tissue inhibitors of MMPs such as TIMP-3 is frequently repressed [176,177,178]. Two different p53 mutants (R248W and D281G) have been shown to inhibit TIMP-3 transcription [176]. In this study, it was demonstrated that overexpression of either of the two p53 mutants in human colon carcinoma cells leads to promoter repression of TIMP-3, resulting in reduced expression and elevated MMP activity [176]. This ultimately causes increased degradation of the extracellular matrix and basal lamina associated with increased metastatic potential. Interestingly, although wild-type p53 may also repress TIMP-3, the physiologically low levels of wild-type p53 in tissues probably allows TIMP-3 to remain highly expressed [176]. Accordingly, p53 mutant proteins are stabilized and accumulate at very high levels in human tumors, reducing TIMP-3 expression and promoting ECM turnover [176]. Of note, direct binding of mutant p53 to the TIMP-3 promoter was not observed [176], suggesting an indirect mechanism of promoter regulation that remains to be investigated. 

Furthermore, mutant p53 constitutively interacts with PML, unlike the transient stress-induced association of PML with wild-type p53 [179,180]. Importantly, PML facilitates mutant p53 to aberrantly activate genes, possibly in the context of the hijacked transcription factor NF-Y [179,180]. In support of a functional interplay, oncogenic functions of mutant p53 are attenuated by PML depletion, and the tumor profile of mice with p53 mutations is shifted when PML is inactivated [179,181]. The exact role of the mutp53/PML axis for the tumor cell secretome, however, remains to be delineated.

### 4.4. Mutant p53 Modifies Exosomal Protein Secretion

Apart from stimulating the expression of secreted proteins or proteins involved in the classical secretory pathway, mutant p53 also exerts effects on protein secretion via extracellular vesicles (EV), comprising plasma membrane-derived “ectosomes” (microparticles/microvesicles) and “exosomes” of endosomal origin (Figure 5) [182]. Exosomes are small (30–200 nm in diameter), single-membrane organelles secreted by cells and enriched in DNA, coding and noncoding RNAs, proteins, and lipids [183,184]. Exosomes are released from a variety of cells, including fibroblasts and tumors, and their activities are diverse, ranging from remodeling of the ECM to cell-cell communication. Accordingly, they are involved in numerous processes, including angiogenesis, apoptosis, antigen presentation, inflammation, and cancer [183,184]. The production of exosomes, as well as their molecular cargo, is affected by external stress signals, such as oxidative stress or ionizing radiation. Being a critical cellular stress-responsive transcription factor, it comes as no surprise that wild-type p53 also plays major roles in exosome biogenesis by regulating both secretion and cargo [185,186]. For instance, wild-type p53 enhances exosome production in response to DNA-damage via transcriptional upregulation of TSAP-6 and CHMP4C [185,186]. In addition, p53 alters the exosomal content and size in colorectal carcinoma cells via HGS, a key component of the endosomal sorting complex [187]. Intriguingly, loading of EVs with anti-metastatic cargo, such as metalloproteinase inhibitor TIMP-3, was shown to be dependent on p53 acetylation by the BAG6/CBP/p300 complex followed by recruitment of the ESCRT (endosomal sorting complex required for transport) machinery [187]. 

While evidence for the physiological role of p53 in exosome biology is accumulating, we are also learning more and more on the multifaceted role of mutant p53 in the regulation of exosomes and the resulting impact on cancer pathogenesis. Obviously, *TP53* gene mutations compromise or ablate the effects of wild-type p53 on exosome biogenesis, as summarized above. In addition, similar as observed for the transcriptional activities of wild-type and mutant p53, mutant p53 proteins acquire novel exosomal functions that are distinct from those of wild-type p53, counteract wild-type p53-mediated tumor suppression, and typically promote tumor progression and metastatic spread. For example, p53-mutant colorectal tumor cells selectively shed miR-1246-enriched exosomes, even though intracellular miR-1246 levels are independent of p53 status [188]. In the model proposed, mutant p53 enhances SUMOylation of the RNA-binding protein hnRNPa2b1, which is required for binding and sorting miR-1246 into exosomes [188]. Uptake of miR-1246-enriched exosomes by neighboring macrophages triggers their reprogramming into a tumor-promoting M2-like state [188]. These tumor-associated macrophages (TAMs) are a chief component of the vast majority of solid tumors and strategic players in cancer progression, as they are responsible for constructing an immunosuppressive and pro-metastatic microenvironment through the production of chemokines, cytokines, and growth factors [188,189]. Although it is still unknown whether other stromal cells are also affected, this study revealed an intriguing cell nonautonomous pro-tumorigenic role of mutant p53 in microenvironmental reprogramming mediated by altered cargo loading of exosomes.

Similarly, a comparative proteomics analysis of p53-mutant and p53-null exosomes revealed a mutp53-associated reduction of podocalyxin (PODXL), a sialomucin associated with cancer aggressiveness [190,191]. In the proposed model, mutant p53 controls PODXL transcription via inhibition of p63 and Rab35 GTPase, which interacts with podocalyxin to influence its sorting to exosomes. When exosomes from p53-mutated cancer cells are taken up by fibroblasts, this promotes trafficking of integrins via the RCP/DGKα pathway. RCP-dependent integrin recycling is a well-characterized GOF mechanism of mutant p53 that increases tumor cell invasive migration in a cell-autonomous manner [123]. In addition, ECM proteins, also within tumors, are continuously assembled and remodeled, and integrin trafficking via the endosomal system can modulate the properties of the ECM [192,193]. Reprogramming of integrin trafficking in fibroblasts through the transfer of mutp53-exosomes therefore results in ECM remodeling and creation of pre-metastatic niches that invite metastatic invasion by tumor cells [191]. 

p53 mutants therefore elegantly modify secretion of extracellular vesicles in multiple ways to shape the cellular and extracellular microenvironment, both at the tumor site and in distant organs, to support invasive growth and metastatic spreading.

## 5. Concluding Remarks

*TP53* mutations are well-known to shape an aggressive and metastatic tumor cell phenotype by various cell-autonomous mechanisms that have been extensively reviewed elsewhere [23,24,194]. As such, mutant p53 is recognized as an important driver of the epithelial-mesenchymal transition (EMT) by controlling TGF-β signaling and by regulating expression of various EMT TFs (ZEB1, ZEB2, Snail, and Slug) or EMT-miRNAs (miR-130b, miR-34, and miR-155) [23]. Furthermore, cell-autonomous effects of mutant p53 on receptor tyrosine kinase expression (EGFR, PDGFRβ, and MET); Myo10-mediated integrin transport; spatial regulation of RhoA; RCP-dependent endocytic trafficking; and ENTPD5-mediated N-glycoprotein-folding further increase the migratory and invasive propensity of tumor cells [23].

Apart from these cell-autonomous mechanisms, *TP53* mutations—by controlling the secretion of soluble and vesicle-bound proteins—have widespread non-cell-autonomous effects on the cellular microenvironment, as outlined in this review. The microenvironmental impact of wild-type p53 is blunted by *TP53* mutations, either directly or by dominant-negative action, and reorganized by neomorphic mutant p53 (gain-of-function) properties. 

Of note, many of the secretome alterations driven by *TP53* mutations have been described only for a few (mostly hotspot) *TP53* mutations and for a single cancer type. It therefore remains to be seen how distinct *TP53* mutations differ in their secretory impact but also how tissues or cancer (sub)types vary in their susceptibility to p53 mutants. For example, p53 mutations were shown to modulate colitis-associated colon cancer in mice through a remarkably wide variety of mechanisms involving upregulation of EMT, NF-κB, or Stat3 signaling, respectively, possibly reflecting slight differences in experimental protocols that lead to distinct inflammatory conditions [95,138,143]. Such context- and mutation-specific pro-tumorigenic properties of *TP53* mutations have also been previously reported and discussed extensively with respect to the cell-autonomous effects of p53 mutants [19,21,24,25,195]. Given that neomorphic properties of p53 mutants are largely mediated through interactions with other proteins—and, in particular, transcription factors, which are known to be expressed in a tissue-specific manner to control cellular differentiation states and specialized cell functions—it is not entirely unexpected that the net secretory effect of a *TP53* mutation is strongly dependent on the set of transcription factors and the chromatin landscape, which define a given tissue or disease setting.

With these caveats in mind, this review aimed to highlight the plethora of mechanisms by which a *TP53* mutation interacts with a given tissue or disease context, reorganizes the regulatory network that it encounters, induces characteristic alterations to the tumor cell secretome, and, thereby, remodels the tumor stroma into a less hostile microenvironment that is more supportive for tumor expansion and metastasis. To obtain a unified generalizing picture of the non-cell-autonomous functions of *TP53* mutations and identify putative drug targets for therapeutic intervention, systematic studies comparing different *TP53* mutations in multiple cancer types are urgently needed.

In summary, the reach of *TP53* mutations extends well beyond the boundary of the p53-mutated cell into the direct neighborhood and via secretion of exosomes even into distant organs, where the mutant p53 secretome contributes to the priming of pre-metastatic niches as future sites of metastasis. The non-cell-autonomous effects of *TP53* mutations therefore massively support the numerous cell-intrinsic, pro-metastatic functions of mutant p53 and make *TP53* mutations even more powerful drivers of tumor growth and metastasis. As such, mutant p53 and mutant p53-triggered cell-extrinsic pathways emerge as even more interesting targets for the treatment of highly aggressive cancer types.

## Figures and Tables

**Figure 1 biomolecules-10-00307-f001:**
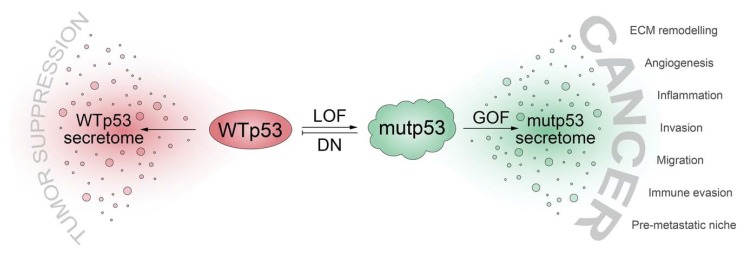
*TP53* mutations shift the cellular secretome from a tumor suppressive secretome driven by wild-type p53 (WTp53) to a cancer-promoting secretome through: loss of WTp53 function (LOF), dominant-negative (DN) inhibition of WTp53 by mutant p53 (mutp53), and gain of neomorphic functions with oncogenic properties (GOF). ECM, extracellular matrix.

**Figure 2 biomolecules-10-00307-f002:**
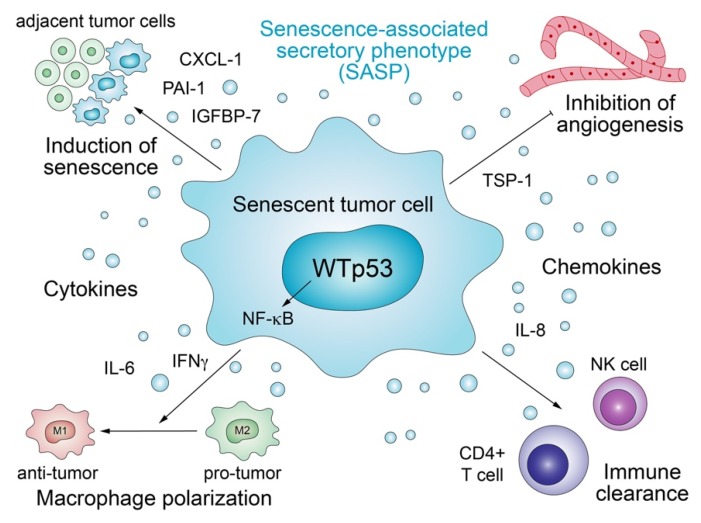
Abrogation of the wild-type p53 secretome by *TP53* mutations. The senescence-associated secretory phenotype (SASP) driven by wild-type p53 induces senescence in neighboring tumor cells, inhibits angiogenesis, polarizes macrophages to the anti-tumor M1 phenotype, and activates NK and T cells to trigger immune clearance.

**Figure 3 biomolecules-10-00307-f003:**
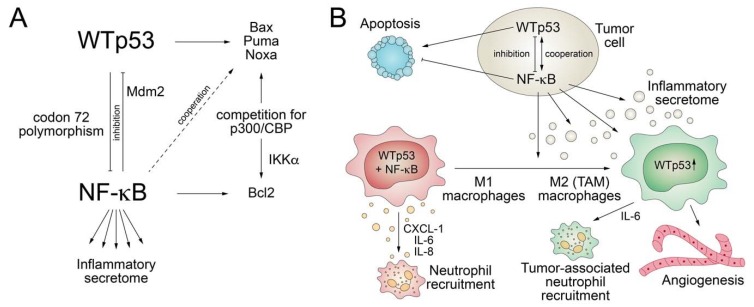
Abrogation of the wild-type p53 secretome by *TP53* mutations. (**A**) Wild-type p53 crosstalks to NF-κB in a mostly inhibitory, but sometimes, also cooperative manner. The outcome is context-dependent manner and modulated by, for example, IKKα and the common *TP53* codon 72 polymorphism. (**B**) p53 crosstalk with NF-κB controls apoptosis and impacts on the inflammatory secretome, thereby regulating macrophage polarization, neutrophil recruitment, and angiogenesis. TAM, tumor-associated macrophages.

**Figure 4 biomolecules-10-00307-f004:**
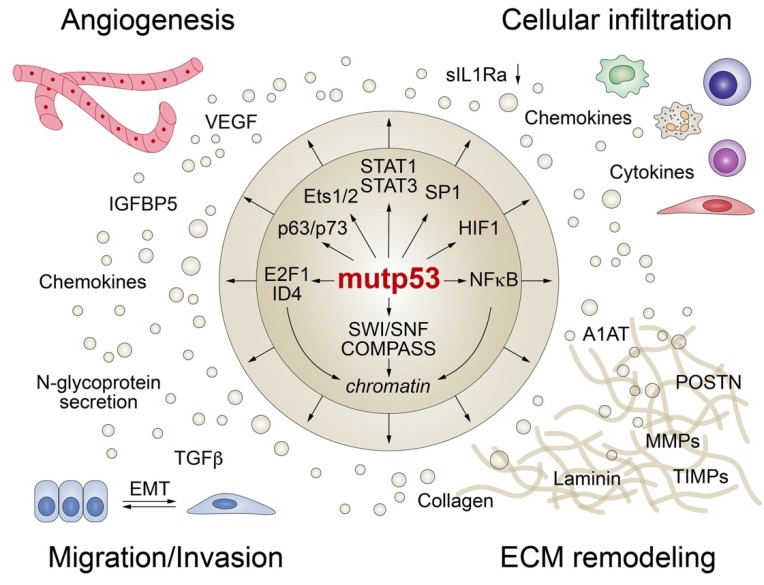
Mutant p53 shapes the tumor cell secretome at the transcriptional level. Mutant p53 interacts with transcription factors and/or chromatin-modifying enzymes (SWI/SNF and COMPASS) to promote angiogenesis, trigger epithelial-mesenchymal transition (EMT), remodel the extracellular matrix (ECM), and modulate the composition and function of the cellular stroma infiltrate.

**Figure 5 biomolecules-10-00307-f005:**
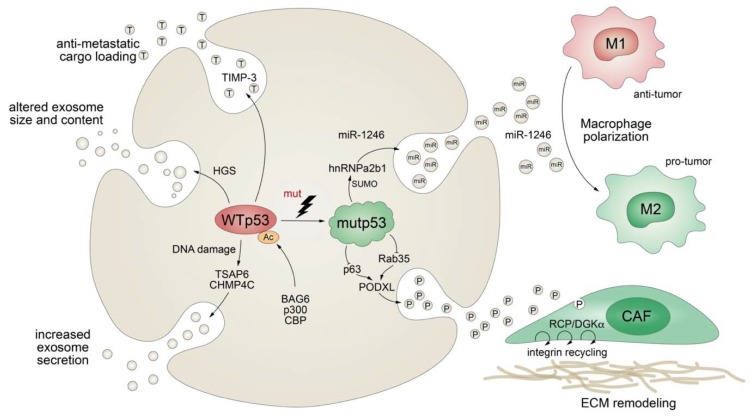
Mutant p53 shapes the tumor cell secretome at the exosome level. *TP53* mutations abrogate wild-type p53-mediated effects on exosome secretion, size, and cargo and actively control exosomal secretion of miR-1246 and podocalyxin (PODXL), thereby affecting macrophage M1-M2 polarization and extracellular matrix (ECM) remodeling by cancer-associated fibroblasts (CAF).

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
