# Peer review of "p53’s Extended Reach: The Mutant p53 Secretome"

_biomolecules, 2020, doi:10.3390/biom10020307_

Round 1

Reviewer 1 Report

Pavlakis and Stiewe have produced a well-written and informative review that will be of significant interest to the field. The manuscript convers an impressive amount of literature, with nearly 200 papers cited, but its logical structure and the use of well-crafted figures makes it an accessible and engaging reading.

The focus of the manuscript is on how p53 mutations alters the activity of the wild type protein or acquire new functions related to the modulation of cancer cell secretomes, leading to the formation of a microenvironment that supports tumor growth and/or spread. The topic, particularly the section on extracellular vesicles, is very relevant and timely.

Here are a number of minor considerations the Authors may want to consider for the final version of the review.

Section 2.1 “Classes of TP53 mutations” is clear and informative, but may be shortened a bit as some of its content does not appear to be directly relevant to the main topic of the review.

Section 2.2 might also be condensed a bit, to focus more readily on setting the stage by introducing figure 1.

Page 4, line 145: some references could be added already at the end of the closing statement of the section.

Section 3.1, lane 173-174: the notion that senescence and SASP can also have pro-tumorigenic effects could be introduced earlier in the paragraph.

Figure 2 panel A: consider if the link between wt p53 and NFkB can be graphically rendered clearer. Line 194: is “and” the best conjunction for that sentence?

Figure 3. Although the paragraph on page 11 is clear and detailed, I wonder if the figure could be improved by adding some details on the links between mutant p53 and chromatin features.

Author Response

Response to Review #1

Pavlakis and Stiewe have produced a well-written and informative review that will be of significant interest to the field. The manuscript convers an impressive amount of literature, with nearly 200 papers cited, but its logical structure and the use of well-crafted figures makes it an accessible and engaging reading.

The focus of the manuscript is on how p53 mutations alters the activity of the wild type protein or acquire new functions related to the modulation of cancer cell secretomes, leading to the formation of a microenvironment that supports tumor growth and/or spread. The topic, particularly the section on extracellular vesicles, is very relevant and timely.

We thank the reviewer for this positive evaluation.

Here are a number of minor considerations the Authors may want to consider for the final version of the review.

Section 2.1 “Classes of TP53 mutations” is clear and informative, but may be shortened a bit as some of its content does not appear to be directly relevant to the main topic of the review.

Section 2.2 might also be condensed a bit, to focus more readily on setting the stage by introducing figure 1.

We have shortened both section 2.1 and 2.2 substantially to focus more on the main topic.

Page 4, line 145: some references could be added already at the end of the closing statement of the section.

We have added two representative references (Line 154).

Section 3.1, lane 173-174: the notion that senescence and SASP can also have pro-tumorigenic effects could be introduced earlier in the paragraph.

Done (Line 173)

Figure 2 panel A: consider if the link between wt p53 and NFkB can be graphically rendered clearer. Line 194: is “and” the best conjunction for that sentence?

We have split the orginial Figure 2B into two separate Figures 3A and 3B. In conjunction with the main text, the new Figure 3A now illustrates the crosstalk between wtp53 and NFkB more clearly and reduces the complexity of Figure 3B, which focuses on the microenvironmental effects of the crosstalk.

Figure 3. Although the paragraph on page 11 is clear and detailed, I wonder if the figure could be improved by adding some details on the links between mutant p53 and chromatin features.

We have slightly changed the figure to illustrate better that mutp53 is altering chromatin through the SWI/SNF and COMPASS complexes.

Reviewer 2 Report

Biomoledules-716634

p53’s extended reach: the mutant p53 secretome

Pavlakis and Stiewe

This is a well-written review that highlights the interplay between tumor cells and the tumor microenvironment composed of different types of immune cells, fibroblasts and angiogenic cells. Many references are provided and the Figures are clearly presented to depict the major features of the review. Any review of p53 is challenging because of the 1) diversity of p53 mutations, 2) the diverse effects of different p53 mutants in many different tumor cell types and 3) the scope of different tumor subtypes within a given tissue, such as the lung prostate or ovary. The new focus of this review based, on the title, is on factors that are released from tumor cells and how they impact tumor behavior and progression. While some of the information is new, what is currently known about exosomes seems a bit limited and the review does not really provide convincing evidence that markers will be found. Often gene abbreviations are provided with a name or function. The review is a bit overwhelming and does not indicate which of the genes or pathways are most relevant to the most common tumor types or subtypes that are present in each tumor type (mesenchymal, epithelial, vascular, etc.

It might be useful to summarize the complexity of p53 actions in one or two tumor types or subtypes to demonstrate the specificities involved for one GOF mutant in the lung versus the prostate. The general figures may not apply to all p53 driven tumors.

Author Response

Response to Review #2

This is a well-written review that highlights the interplay between tumor cells and the tumor microenvironment composed of different types of immune cells, fibroblasts and angiogenic cells. Many references are provided and the Figures are clearly presented to depict the major features of the review.

We thank the reviewer for this positive evaluation.

Any review of p53 is challenging because of the 1) diversity of p53 mutations, 2) the diverse effects of different p53 mutants in many different tumor cell types and 3) the scope of different tumor subtypes within a given tissue, such as the lung prostate or ovary. The new focus of this review based, on the title, is on factors that are released from tumor cells and how they impact tumor behavior and progression. While some of the information is new, what is currently known about exosomes seems a bit limited and the review does not really provide convincing evidence that markers will be found.

We agree that there is much less information on exosomes than soluble factors as mutant p53 effectors. However, exosome research is a rather new and still emerging field. Given that reviewer #1 specifically highlighted this section as important and timely, we decided to keep this section in the review.

Often gene abbreviations are provided with a name or function. The review is a bit overwhelming and does not indicate which of the genes or pathways are most relevant to the most common tumor types or subtypes that are present in each tumor type (mesenchymal, epithelial, vascular, etc.). It might be useful to summarize the complexity of p53 actions in one or two tumor types or subtypes to demonstrate the specificities involved for one GOF mutant in the lung versus the prostate. The general figures may not apply to all p53 driven tumors.

We believe that the reviewer raises a very important point. Much of what has been described for mutant p53 has only been observed in a limited number of cancer types and for a subset of all p53 mutations. At present it is unclear, which secretory effects of mutant p53 are more general and which are specific for certain mutations and cancer (sub)types. Unfortunately, systematic studies into this important topic are missing. Throughout the review we have therefore mentioned that the observations only hold true for "certain" or "some" mutants and have in many places indicated precisely in which tissues or cancer types the effects were observed. For more details on the exact mutants and tissues that were studied, the reader is referred to the references to maintain a reasonable length of the review in total. However, we have added two paragraphs to the section 5 „Concluding remarks“ that explicitely addresses these limitations, discusses possible causes and emphasizes the importance of further systematic studies into these issues.